# *PAC-tuning*: Fine-tuning Pretrained Language Models with PAC-driven Perturbed Gradient Descent

Guangliang Liu[*], Zhiyu Xue[+], Xitong Zhang[*], Kristen Marie Johnson[*] and Rongrong Wang[*]

[*]Michigan State University
[+] UC Santa Barbara
{liuguan5,zhangxit,kristenj,wangron6}@msu.edu
zhiyuxue@ucsb.edu

## Abstract

Fine-tuning pretrained language models (PLMs) for downstream tasks is a large-scale optimization problem, in which the choice of the training algorithm critically determines how well the trained model can generalize to unseen test data, especially in the context of few-shot learning. To achieve good generalization performance and avoid overfitting, techniques such as data augmentation and pruning are often applied. However, adding these regularizations necessitates heavy tuning of the hyperparameters of optimization algorithms, such as the popular Adam optimizer. In this paper, we propose a two-stage fine-tuning method, PAC-tuning, to address this optimization challenge. First, based on PAC-Bayes training, PAC-tuning directly minimizes the PAC-Bayes generalization bound to learn proper parameter distribution. Second, PAC-tuning modifies the gradient by injecting noise with the variance learned in the first stage into the model parameters during training, resulting in a variant of perturbed gradient descent (PGD). In the past, the few-shot scenario posed difficulties for PAC-Bayes training because the PAC-Bayes bound, when applied to large models with limited training data, might not be stringent. Our experimental results across 5 GLUE benchmark tasks demonstrate that PAC-tuning[1] successfully handles the challenges of fine-tuning tasks and outperforms strong baseline methods by a visible margin, further confirming the potential to apply PAC training for any other settings where the Adam optimizer is currently used for training.

## 1 Introduction

Since the emergence of pretrained language models (PLMs), e.g., BERT (Devlin et al., 2018) and GPT-3 (Brown et al., 2020), fine-tuning of such pretrained models has been the *de-facto* pipeline for NLP, achieving state-of-the-art results in various tasks. There are two main fine-tuning approaches: parameter-tuning and prompt-tuning. Parameter-tuning considers a PLM as a feature extractor and tries to update the complete PLM with a small learning step (Jiang et al., 2020; Gunel et al., 2020). By contrast, prompt-tuning aligns downstream tasks with the objective of language modeling through inserting prompts with or without task demonstrations into the original sample and asking the PLM to predict the next token according to the prompted input context (Gao et al., 2021; Gu et al., 2022; Hu et al., 2022b). In the context of few-shot learning, parameter-tuning over a neural model with up to billions of parameters is a non-trivial task (Zhang et al., 2020; Dodge et al., 2020). A significant challenge is that the training process is unstable (Mosbach et al., 2020; Lee et al., 2019): given only a few samples from downstream tasks, the overparameterization nature of a PLM leads to issues such as overfitting and forgetting (He et al., 2021; Kirkpatrick et al., 2017). Existing methods mainly address these challenges by applying data augmentation (Zhou et al., 2022; Wu et al., 2022; Kumar et al., 2019), regularization (Zhu et al., 2019; Yu et al., 2021; Aghajanyan et al., 2020; Jiang et al., 2020) and network pruning (Xu et al., 2021).

From the perspective of machine learning theory, data augmentation, regularization, and pruning are all used during training as *generalization enhancers*. Other well-known generalization enhancers include weight-decay and dropout. Perhaps a little surprisingly, different choices of learning rates (Li et al., 2019) and mini-batch sizes (He et al., 2019) also affect generalization. A less wellknown enhancer to the NLP community is noise injection, implemented by the PGD (Perturbed Gradient Descent) algorithm (Orvieto et al., 2022, 2023). Theoretically, PGD is shown to effectively help the algorithm escape spurious local minima (Zhou et al., 2019) and saddle points (Jin et al., 2021),

---

[1]Our implementation is publicly available at https://github.com/MSU-NLP-CSS/PAC-tuning

due to an implicit regularization on the trace of the Hessian matrix.

In this paper, instead of searching for the optimal combination of a basket of generalization enhancers, we follow an alternative training framework, PAC-Bayes training (Rivasplata et al., 2019), which provides a more straightforward way of improving the generalization – the network is trained towards directly minimizing the generalization error characterized by the PAC-Bayes bound (Maurer, 2004) instead of minimizing only the training loss. Although PAC-Bayes bounds are classical bounds in learning theory, leveraging them for training is relatively new. This is likely due to concerns that the PAC-Bayes bounds suffer from the curse of dimensionality (Dziugaite and Roy, 2017a; Foong et al., 2021); therefore they are unlikely to be effective on modern neural networks that are large and deep. However, recent studies (Rivasplata et al., 2019; Zhang et al., 2023) have shown that this view might be too pessimistic, and that the PAC-Bayes bound could be rather effective in training modern convolutional neural networks.

This paper explores the potential of PAC-Bayes training on even larger models, namely, the PLM. We consider the most challenging task in the perspective of PAC-Bayes training, the fine-tuning task, as it amounts to using an extremely small training dataset to tune a PLM with millions of parameters. For this setting, we propose a novel, efficient implementation of PAC-Bayes training, called PAC-tuning, which consists of two stages. The first stage learns the noise variance and updates the PLM's parameters by minimizing a PAC-Bayes upper bound. The second stage implements noise injection training with the noise variance learned from the previous stage. We validate the effectiveness of PAC-tuning with few-shot text classification tasks extracted from the GLUE benchmark. The overall good performance of PAC-tuning suggests promising potential for leveraging PAC-Bayes training for fine-tuning much larger PLMs, even during the pretraining process. To the best of our knowledge, PAC-tuning is the first work of its kind in terms of improving PLM fine-tuning by PAC-Bayes training.

## 2 Related Works

**Few-shot learning with PLMs** has been comprehensively studied by Zhang et al. (2020) to understand the influence of various factors, such as the layer-wise learning rate and instability of cross-entropy loss, in order to recommend techniques for improving the final generalization performance. In contrast to fine-tuning methods which require an update for the model parameters, another research line explores prompting-based methods; Prefix-tuning (Li and Liang, 2021) is a representative one. A straightforward solution for few-shot tasks is to generate more data via data augmentation (Arthaud et al., 2021; Feng et al., 2021). To grapple with the forgetting issue of fine-tuning PLMs, trust-region-based methods define a trustworthy region constraining the change of parameters in each update step. Based on the lottery-ticket hypothesis behind PLMs (Frankle and Carbin, 2018), parameter-tuning methods that only update the sub-network of a PLM have also been proposed (Ben Zaken et al., 2022). All these methods require heavy hyperparameter searches and minimization of the training error, instead of directly optimizing over the generalization error.

**PAC-Bayes Training** means training machine learning models by minimizing the PAC-Bayes upper bound. In contrast to empirical risk minimization, PAC-Bayes training is more straightforward in terms of improving generalization by minimizing the upper bound of generalization error. McAllester (1998) trains a stochastic neural network on an MNIST dataset by minimizing a non-vacuous PAC-Bayes bound. The PAC-Bayes training with BackProp proposed by Rivasplata et al. (2019) trains shallow probabilistic neural networks and certifies their risk by PAC-Bayes bound on the MNIST dataset. Zhang et al. (2023) introduced Auto-tune PAC to train various neural networks, such as ResNet and GNN, through optimizing both the prior distribution variance and posterior distribution variance of parameters. Auto-tune PAC leverages a larger model and larger dataset, including ResNet34, DenseNet121, and the CIFAR 100 dataset, and the authors test a GNN on a smaller dataset with only 20 nodes per class. Previous works overlook confidence difference between pretrained layers and adaptation layers, this is the main reason that those works can not be applied to PLMs. We, however, take the confidence difference into account by learning the noise level associated with pretrained layers and adaptation layers separately.

**Perturbed Gradient Descent (PGD)** implicitly regularizes the trace of the Hessian matrix to push the model towards a region of the loss land-

scape with larger flatness, which is claimed to be a measurement of generalization (Foret et al., 2020; Jiang et al., 2019). Zhou et al. (2019) proves that PGD can help a two-layer convolutional neural network model escape a spurious local minimum and converge to a global minimum. A similar generalization-enhanced benefit of PGD is also validated by Jin et al. (2021): given PGD, neural network models can converge to second-order stationary points and avoid saddle points. While existing PGD works assign isotropic noise to models, causing training loss explosion, PAC-tuning avoids this problem by injecting parameter-wise noises to PLMs.

## 3 Method

This section presents our proposed method, PAC-tuning, an implementation of PAC-Bayes training for parameter-based fine-tuning of PLMs. Section 3.2 introduces PAC-Bayes training and the PAC-Bayes bound, followed by Section 3.3 which describes perturbed gradient descent. The motivation to assist PGD with PAC-Bayes training is presented in Section 3.4, and we explain the details of PAC-tuning in Section 3.5.

### 3.1 Problem Setup and Notations

Let $\theta$ be the parameters of the PLM. We replace the head layer of the PLM with a one-layer fully-connected neural network parameterized by $\omega$. Denoting the PLM classifier as $f$, we consider $\theta$ and $\omega$ as vectors for simplicity. Let $\ell(\cdot; \theta, \omega)$ be the loss function, e.g., the cross-entropy loss. An individual sample is represented with $(x, y)$ where $x$ is the input data and $y$ is the associated label.

### 3.2 PAC-Bayes Training and the PAC-Bayes Bound

The idea of PAC-Bayes training arises from minimizing the PAC-Bayes bound $J(\theta, \mathcal{Q}, \mathcal{P}) \equiv L_{\text{train}} + L_{\text{PAC}}$ of the following type:

$$
\overbrace{\mathbb{E}_{\theta \sim \mathcal{Q}} \mathbb{E}_{(x,y) \sim \mathcal{D}} \ell(x, y; \theta)}^{\text{generalization error}}
$$
$$
\leq \underbrace{\frac{1}{m} \sum_{i=1}^{m} \mathbb{E}_{\theta \sim \mathcal{Q}} \ell(x_i, y_i, \theta)}_{L_{\text{train}}} + \underbrace{\sqrt{\frac{\log \frac{1}{\delta} + \text{KL}(\mathcal{Q}||\mathcal{P})}{2m}}}_{L_{\text{PAC}}}
$$

PAC-Bayes bounds are probabilistic bounds that hold with high probabilities, i.e., $1 - \delta$ (where $\delta$ is the probability that the upper bound does not

hold), and for any neural network type. They characterize the generalization error of a trained model. Here, $\theta$ is the weight of the neural network, $m$ is the number of training samples, $\mathcal{Q}$ and $\mathcal{P}$ are arbitrary pairs of prior and posterior distributions, KL is the Kullback–Leibler divergence measuring the distance between two distributions, and $\mathcal{D}$ is the training data distribution. When the PAC-Bayes bound is nonvacuous, minimizing the bound effectively reduces the generalization error. In several recent works (Rivasplata et al., 2019; Zhang et al., 2023), optimization algorithms have been proposed to find the minimizer of $J(\theta, \mathcal{Q}, \mathcal{P})$ when $\mathcal{Q}$ and $\mathcal{P}$ are taken to be multivariate Gaussian distributions. This provides an automatic way to learn the optimal noise levels (which are the variance of $\mathcal{Q}$) that reflect the different confidence levels of each parameter in the model $\theta$.

### 3.3 Noise Injection and Perturbed Gradient Descent (PGD)

The KL term in the $L_{\text{PAC}}$ may suffer from two possible issues: (1) it could be difficult to compute and (2) it could be too large to allow the training loss to approach 0. As a result, it is a common practice to ignore the $L_{\text{PAC}}$ term in the PAC-Bayes bound and simply minimize $L_{\text{train}}$. In the simplest case, we use isotropic Gaussian noise, $N(\theta, \eta\mathbf{I})$, with mean $\theta$ and noise level $\eta$ as the posterior distribution, and then $L_{\text{train}}$ reduces to:

$$
L_{\text{train}} = \frac{1}{m} \sum_{i=1}^{m} \mathbb{E}_{\tau \sim N(\mathbf{0}, \mathbf{I})} \ell(x_i, y_i, \theta + \eta\tau) \quad (1)
$$

This can be interpreted as the original training loss with noise injected into the model parameters, and our goal is to minimize its expectation.

The algorithm that minimizes $L_{\text{train}}$ is called Perturbed Gradient Descent (PGD), which injects random noise into the model before computing the gradient and removes the added noise after the gradient update.

Algorithm 1 describes the application of *Perturbed Gradient Descent* to the PLM. Specifically, in line 2, noises $\tau_1$ and $\tau_2$ are sampled from a standard Gaussian distribution whose dimension is the same as $\theta$ and $\omega$ (we refer readers to the confidence difference issue in Section 3.5), respectively. Next, we rescale the sampled noises by $\eta_1$ and $\eta_2$ and inject them into the parameters of the PLM $f$ to produce noisy parameters $\theta'$ and $\omega'$ as shown in line 3. Parameters are then updated according to

**Algorithm 1:** Perturbed Gradient Descent

---
1 Sample $(x, y)$ from training dataset
2 Sample noise from a Gaussian distribution
   $\tau_1, \tau_2 \sim N(\mathbf{0}, \mathbf{I})$
3 Rescale and inject noise
   $\theta', \omega' = \theta + \sqrt{\eta_1} \cdot \tau_1, \omega + \sqrt{\eta_2} \cdot \tau_2$
4 Update parameters $\theta, \omega =$
   $\theta - \alpha_1 \cdot \frac{\partial L(x; \theta', \omega')}{\partial \theta}, \omega - \alpha_2 \cdot \frac{\partial L(x; \theta', \omega')}{\partial \omega}$

---

the perturbed gradient with a learning rate of $\alpha_1$ and $\alpha_2$, as shown in line 4.

### 3.4 The Noise Level

In the previous section, we explained why $L_{\text{train}}$ amounts to a noise injection into the model. Next, we provide the intuition of why the proposed algorithm can detect the noise variance automatically. When we introduce noise into the model, the training loss $L_{\text{train}}$ is expected to rise. The greater the amount of noise added, the larger the anticipated increase in $L_{\text{train}}$. In other words, $L_{\text{train}}$ is generally an increasing function of the noise level. Hence if we just minimize $L_{\text{train}}$, then the optimal noise variance would just be 0. The reason our algorithm can learn a meaningful non-zero noise is due to the existence of the second term $L_{\text{PAC}}$ in the loss, which is a decreasing function of the noise level when the noise converges towards the prior distribution. As a result, we expect that minimizing the total loss, $L_{\text{train}} + L_{\text{PAC}}$, will find us an optimal point for the noise level, and therefore achieve automatic learning of the noise. This is the basic idea of the proposed PAC-tuning algorithm that will be described in the next section.

After the training is complete, the learned noise levels can be used for model interpretation/validation, as they reflect how important each model parameter is to the final performance. For example, a model parameter associated with a large learned noise level is less important than one with a small noise level. More concretely, if the trained model parameter is $(1, 1, 1)$ and the learned noise level by PAC-training is $(10, 1, 10)$, then it indicates that the second model parameter is more important than the first and the third because its associated noise injection level is low.

### 3.5 PAC-tuning

Previous work on PAC-Bayes training all targeted the one-time training of a neural network. In fine-tuning, we train the model a second time, and therefore we expect the pretrained part to be updated less in the second round. In other words, $\theta$ should not change much since the PLM is assumed to be accurate enough and we generally use a small learning rate to update $\theta$ (Zhang et al., 2021). In contrast, the learning rate for $\omega$ should be much larger because we are less confident about it. We name this as the *confidence difference issue*. Recall in Section 3.4, we explained how the noise level reflects our confidence in the target parameters. Therefore, we are motivated to use different noise levels as well as learning rates for $\theta$ and $\omega$. In turn, the KL term in the $L_{\text{PAC}}$ would consist of two parts:

$$\text{KL}(\mathcal{Q}_\omega || \mathcal{P}_\omega) + \text{KL}(\mathcal{Q}_\theta || \mathcal{P}_\theta)$$

To force these KL divergences small for extremely large models, we leverage the PAC-Bayes bound proposed in Zhang et al. (2023), a variant of the basic PAC-Bayes bound $J(\theta, \mathcal{P}, \mathcal{Q})$ described in Section 3.2. The final objective function we want to minimize, omitting learnable parameters of the prior distribution variance, e.g., $\lambda$ and $\beta$, for simplicity, is $J(\cdot; \xi, \epsilon, \theta, \omega)$:

$$J(D; \xi, \epsilon, \theta, \omega) = \overbrace{\frac{1}{m} \sum_i^m \ell(x_i, y_i; \theta, \omega)}^{L_{\text{train}}}$$
$$+ \underbrace{\frac{(\ln \frac{1}{\delta} + \text{KL}(\mathcal{Q}_\xi^\theta || \mathcal{P}_\lambda^\theta) + \text{KL}(\mathcal{Q}_\epsilon^\omega || \mathcal{P}_\beta^\omega))}{\gamma m} + \gamma K^2}_{L_{\text{PAC}}}$$

where $\xi$ and $\epsilon$ are the posterior distribution variance associated with $\theta$ and $\omega$ respectively, $D = \{(x_i, y_i)\}_{i=1}^m$ is the training dataset, $\delta \in (0, 1)$ is the probability of failure, $\gamma$ can be set to any value within a bounded $[\gamma_1, \gamma_2]$ specified by the users, and $K(\lambda, \beta, \gamma_1, \gamma_2) > 0$ is the effective variance of the training loss $\ell$ when the prior variances for $(\theta, \omega)$ are set to $(\lambda, \beta)$. We refer readers to Section 4 of Zhang et al. (2023) for more details about $\gamma$ and $K$. This objective function is obtained by making the following assumptions: (1) the prior distributions of the PLM classifier are $\mathcal{P}_\lambda^\theta = N(\theta_0, \lambda \mathbf{I})$ and $\mathcal{P}_\beta^\omega = N(\omega_0, \beta \mathbf{I})$, where $\theta_0$ and $\omega_0$ are the initialized parameter weights, and (2) in each gradient update $t$, the posterior distributions of the PLM classifier are $\mathcal{Q}_\xi^\theta = \theta_t + N(\mathbf{0}, \text{diag}(\xi))$ and $\mathcal{Q}_\epsilon^\omega = \omega_t + N(\mathbf{0}, \text{diag}(\epsilon))$ where $\theta_t$ and $\omega_t$ are the current parameter weights for the gradient update step $t$.

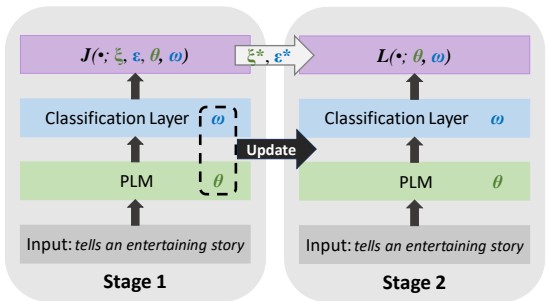

Figure 1: PAC-tuning Pipeline. In *Stage 1*, we update the model parameters and noise variance by minimizing the PAC-Bayes bound $J$ described in Section 3.2. Then the optimal noise variance $\xi^*$ and $\epsilon^*$ are learned after $T_1$ training epochs. Next, fine-tuning is continued in *Stage 2* with Algorithm 1 but the noise variance is fixed as $\xi^*$ and $\epsilon^*$, and the objective function is training loss $L_{\text{train}}$ only. Model parameters $\theta$ and $\omega$ are updated during both fine-tuning stages.

Figure 1 shows the pipeline of our proposed PAC-tuning technique. The implementation contains two stages. In Stage 1, by minimizing the objective $J$ over $T_1$ epochs, the optimal noise variance $\xi^*$ and $\epsilon^*$ and the model parameters $\theta$ and $\omega$ are updated. Afterward, we leverage PGD on the PLM (Algorithm 1) with fixed noise levels as $\xi^*$ and $\epsilon^*$ to update $\theta$ and $\omega$. Two stages of fine-tuning are required because minimizing $J$ cannot usually make the PLM classifier fit the downstream data very well due to the existence of the $L_{\text{PAC}}$ term. During stage 2, the $L_{\text{PAC}}$ term is dropped; therefore the PLM classifier can fit the downstream data well.

The target of **Stage 1** is to estimate posterior variance $\xi$ and $\epsilon$ as well as update model parameters[2]:

$$\xi^*, \epsilon^*, \theta^*, \omega^* = \underset{\xi, \epsilon, \theta, \omega}{\operatorname{argmin}} J(D; \xi, \epsilon, \theta, \omega)$$

To reflect our greater confidence in $\theta$ than $\omega$, we initialize $\xi$ to be smaller than $\epsilon$. Meanwhile, we follow the convention to use a smaller learning rate for the $\theta$ than $\omega$. The small learning rate of $\theta$ would in turn result in a smaller gradient for the corresponding noise $\xi$. Therefore, we need a larger learning rate for $\xi$ to neutralize this effect. In addition, dropout introduces extra noise to model parameters (Wei et al., 2020), resulting in a conflict with the noise injection by our proposed method. To effectively employ PAC-tuning, dropout must be disabled.

Given the learned posterior variance, **Stage 2** continues fine-tuning $\theta$ and $\omega$ through PGD. In

---

[2]Please note $\lambda$ and $\beta$ are also optimized in Stage 1.

each gradient update, we sample noise $\tau_1$ and $\tau_2$ from a standard normal distribution and multiply them by the learned noise variance ($\xi^*$ and $\epsilon^*$) from Stage 1 to replace line 3 of Algorithm 1 as:

$$\theta' = \theta + \sqrt{\xi^*} \cdot \tau_1$$
$$\omega' = \omega + \sqrt{\epsilon^*} \cdot \tau_2$$

## 4 Experiments and Analysis

In this section, we outline our experimental settings, dataset, and baseline models used for evaluation. Section 4.4 discusses the experimental findings. Section 4.5 concludes with an analysis of the stability of our PAC-tuning approach.

### 4.1 Experimental Settings

We conduct extensive experiments with PAC-tuning and baseline PLMs over 5 text classification tasks of the GLUE benchmark [3] as shown in Tables 1 and 2. We adopt the HuggingFace implementations of BERT [4] and GPT-2 [5] as a backbone and add one fully-connected layer to be the classification layer. To simulate a few-shot learning scenario, we randomly sample 100 instances from the original training set and use the whole development set to evaluate the classification performance. All experiments are repeated 5 times and we report the average performance over 5 seeds[6] on the original development set. All model architectures have the same hyperparameters and optimizers in all experiments, except the training epochs in PAC-tuning (as further detailed in Appendix A). We freeze the parameters associated with embeddings and do not update them during fine-tuning.

For the implementation of PAC-tuning, we set the learning rate for the variances associated with the PLM, $\xi$ and $\lambda$, to 0.1, and the learning rate for the variances of the classification layers, $\epsilon$ and $\beta$, to be initialized as 0.5 and decreased to be 90% every 10 gradient updates, until the minimal of 0.01. We chose the loss interval $\gamma$ as 10 for the tasks of SST and CoLA, and used 5 for the remaining tasks. PAC-tuning Stage 1 runs for 250 epochs with a maximum training epoch of 300. However, the convergence of Stage 1 depends on the difficulty of the considered task. For the SST task, a stage 1 with 100 epochs can ensure convergence,

---

[3]https://gluebenchmark.com/
[4]https://huggingface.co/bert-base-uncased
[5]https://huggingface.co/gpt2
[6]Seeds used: 1, 2, 10, 26, 100

but 250 epochs of Stage 1 is enough for all of the experiments reported in this paper.

## 4.2 Dataset

Five tasks of the GLUE benchmark are used to validate our proposed fine-tuning method: the Corpus of Linguistic Acceptability (CoLA), the Stanford Sentiment Treebank (SST), a mixture of the two datasets MultiNLI Matched and MultiNLI Mismatched (MNLI-m), Question NLI (QNLI), and Recognizing Textual Entailment (RTE).

## 4.3 Baseline Methods

The following baseline methods represent current, typical approaches for fine-tuning.

- **Vanilla-tuning** is the vanilla, basic parameter-tuning without any add-on regularization.
- **Data Augmentation** is implemented in this work with BackTranslation (Sennrich et al., 2016) to control the quality of augmented data. BackTranslation is a model-based augmentation method, which first translates a sequence of tokens into another language and then translates it back to the original language. We mix the sampled training data and augmentation data together as the training set. For benchmarks with paired inputs, e.g., MNLI-m, QNLI, and RTE, we generate 2 augmented samples for each training sample. One is generated from the first part of the input and the other is generated using the second part of that input. For the remaining benchmarks (SST and CoLA), we generated only one augmented sample using back translation.
- **Noise Injection** (Orvieto et al., 2023) theoretically and empirically proves that noise injection into a randomly selected layer in each gradient update can avoid large loss variance and effectively implement explicit regularization to overparameterized models.
- **Low-Rank Adaptation (LoRA)** (Hu et al., 2022a) aims to address the challenges of fine-tuning PLMs by leveraging low-rank approximations of the model's weight matrices, achieving more efficient adaptation to specific tasks or domains. The low-rank adaptation matrix amplifies important features for specific downstream tasks that were learned but not emphasized in the general pretrained model, making the adaptation process more

efficient while alleviating overfitting to downstream tasks.

- **Prefix-tuning** (Li and Liang, 2021; Liu et al., 2022) optimizes a sequence of continuous task-specific vectors added to the beginning of the input sequence, known as prefixes, while keeping the PLM parameters frozen. It provides a more efficient and effective approach for fine-tuning PLMs by optimizing these continuous prefixes.
- **BitFit** (Ben Zaken et al., 2022) is a subnetwork fine-tuning method that only optimizes the bias terms of the PLM. By targeting a specific subset of the model parameters, BitFit achieves competitive performance by fine-tuning the entire model, and is especially effective with smaller training datasets.

## 4.4 Experimental Results

Tables 1 and 2 show the experimental results of our proposed PAC-tuning approach compared to other fine-tuning methods when used with the two backbone PLMs, BERT and GPT-2, respectively. The first column lists the specific fine-tuning method as described in Section 4.3. The first three techniques, vanilla-tuning, data augmentation, and noise injection are instances of parameter-tuning methods. The next two techniques, LoRA and prefix-tuning, are examples of parameter-efficient-tuning. The next 5 columns correspond to each GLUE benchmark task. Results for each task are reported in terms of accuracy, except for the CoLA task which uses the Matthew's correlation coefficient (MCC). The final column reports the average results for each fine-tuning approach across all 5 tasks.

Overall, PAC-tuning achieves the best average performance with both PLMs, but is not the best fine-tuning approach for the MNLI-m task given the BERT-base backbone. The average performance of parameter-tuning methods is better than that of parameter-efficient tuning methods, though LoRA is the second best fine-tuning method in Table 1. These experimental results show supportive evidence for future research in applying PAC-tuning for fine-tuning PLMs for downstream tasks.

When applied to the BERT backbone (Table 1), in the tasks of CoLA and SST, PAC-tuning's performance exceeds other fine-tuning baselines by a large margin. The performance gain for the QNLI and RTE tasks is somewhat smaller, but still significant. However, PAC-tuning is worse than data aug-

| BERT | CoLA (MCC) | SST (Accuracy) | MNLI-m (Accuracy) | QNLI (Accuracy) | RTE (Accuracy) | Average |
|---|---|---|---|---|---|---|
| Vanilla-tuning | .235 | .773 | .369 | .702 | .589 | .533 |
| Data Augmentation | .171 | .817 | **.395** | .705 | .594 | .536 |
| Noise Injection | .233 | .783 | .371 | .706 | .588 | .536 |
| LoRA | .298 | .792 | .385 | .669 | .592 | .547 |
| Prefix-tuning | .191 | .704 | .375 | .649 | .565 | .497 |
| BitFit | .267 | .768 | .376 | .647 | .588 | .510 |
| PAC-tuning | **.335** | **.834** | .387 | **.709** | **.601** | **.573** |

Table 1: Experimental Results for BERT-base-uncased Backbone Model. The best results are highlighted in bold. PAC-tuning outperforms other fine-tuning methods in all tasks except MNLI-m, where data augmentation contributes to the best performance. PAC-tuning achieved the best average performance across all 5 tasks.

| GPT-2 | CoLA (MCC) | SST (Accuracy) | MNLI-m (Accuracy) | QNLI (Accuracy) | RTE (Accuracy) | Average |
|---|---|---|---|---|---|---|
| Vanilla-tuning | .048 | .774 | .353 | .560 | .572 | .461 |
| Data Augmentation | .031 | .777 | .355 | .573 | .574 | .462 |
| Noise Injection | .073 | .713 | .369 | .550 | .552 | .451 |
| LoRA | .053 | .703 | .366 | .545 | .560 | .445 |
| Prefix-tuning | .071 | .521 | .352 | .523 | .525 | .398 |
| BitFit | .047 | .586 | .366 | .542 | .520 | .432 |
| PAC-tuning | **.085** | **.815** | **.373** | **.576** | **.580** | **.486** |

Table 2: Experimental Results for GPT-2 Backbone Model. The best results are highlighted in bold. PAC-tuning outperforms all other fine-tuning methods.

mentation and parameter-efficient tuning methods for the MNLI-m task. Data augmentation-based fine-tuning is the best method for the task of MNLI-m and is the second or third best method for SST, QNLI, and RTE. However, it performs worse than vanilla-tuning for the CoLA task. It is the second best method in terms of stable performance across five tasks, indicating the effectiveness of data augmentation in a low-resource setting.

According to Table 2, the overall performance of GPT-2 is worse than that of BERT-based fine-tuning methods, particularly in the task of CoLA. This is consistent with previous findings (Liu et al., 2021; Radford et al., 2019). However, the addition of our method improves the fine-tuned performance, and our method is the best fine-tuning approach for all tasks. All fine-tuning methods show similar trends in performance as with the BERT backbone.

The overall good performance of PAC-tuning proves its feasibility and usefulness for fine-tuning PLMs for few-shot text classification tasks. This typical application scenario introduces two key challenges for applying PAC-Bayes training: larger model sizes and smaller data sizes, which are generally considered to result in vacuous bounds, pre-

venting the use of PAC-Bayes training in practical settings. Our results of PAC-tuning demonstrate that PAC-Bayes training can be used with even very large models like PLMs, that have never been considered before.

## 4.5 Stability Analysis

The PAC-Bayes bound contains a term relevant to data size and the KL-divergence term is associated with model size. Therefore, we conduct thorough experiments to analyze how PAC-tuning's performance changes given different data sizes and model sizes. Table 3 shows the performance of BERT-based fine-tuning methods with respect to a training dataset size of 50 and 20. We construct the training dataset by implementing random sampling over the training set of SST and RTE. When training data size drops to 20, the performance of PAC-tuning is worse than prefix-tuning by a very small margin. Considering the test size of RTE is small, the performance difference between prefix-tuning and PAC-tuning implies that both methods have very close generalization performances.

Table 4 describes the classification results when considering fine-tuning methods on the SST and RTE tasks with BERT-large-uncased as the back-

| SST Training Size | 50 | 20 | RTE Training Size | 50 | 20 |
|---|---|---|---|---|---|
| Vanilla-tuning | .772 | .601 | Vanilla-tuning | .530 | .517 |
| Data Augmentation | .783 | .585 | Data Augmentation | .533 | .507 |
| LoRA | .756 | .596 | LoRA | .538 | .514 |
| Prefix-tuning | .672 | .572 | Prefix-tuning | .538 | **.536** |
| BitFit | .746 | .610 | BitFit | .543 | .517 |
| PAC-tuning | **.810** | **.620** | PAC-tuning | **.546** | .532 |

Table 3: Stability Analysis of Training Dataset Sizes. This table presents the accuracy on development sets for the SST and RTE tasks while varying training dataset sizes. PAC-tuning's performance drops as the CoLA data size decreases to 20 but is still the best fine-tuning method given 50 training samples.

| BERT-large | SST | RTE |
|---|---|---|
| Vanilla-tuning | .832 | .561 |
| Data Augmentation | .836 | **.591** |
| LoRA | .845 | .560 |
| Prefix-tuning | .721 | .542 |
| BitFit | .804 | .546 |
| PAC-tuning | **.848** | .565 |

Table 4: Stability Analysis for a Larger Model Architecture. PAC-tuning outperforms other fine-tuning methods for the task of SST when using BERT-large-uncased. The training data size is fixed at 100.

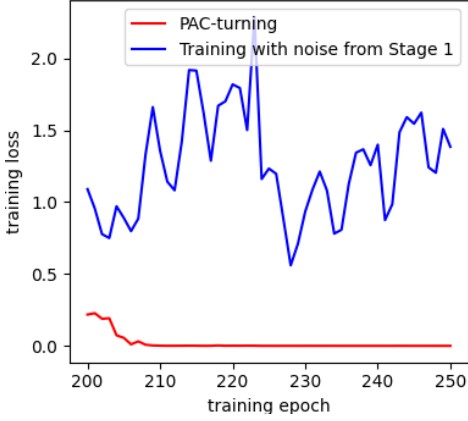

Figure 2: From the beginning of Stage 2, the noise learned in Stage 1 is applied to fine-tune a BERT-based model on the SST dataset from scratch, as the blue line shows. Beginning at the $200^{th}$ epoch we continue PAC-tuning, as shown in the red line, and leverage the learned noise in Stage 1 to fine-tune the model from scratch, as described by the blue line.

bone model. With this larger model, all fine-tuning methods have a performance increase over the two tasks. PAC-tuning is the best method in the SST task and the second best in the task of RTE, where data augmentation outperforms all methods. When viewed with the main experimental results of Section 4.4, these stability tests further validate the usefulness of leveraging PAC-Bayes training, via PAC-tuning, to fine-tune PLMs in the challenging settings of small training data availability and extremely large pretrained models.

## 5 Discussion

### 5.1 The Role of Stage 1

Stage 1 learns the noise variance to be used in Stage 2 and prepares the model to be at a good initialization state for Stage 2. Figure 2 indicates that if we start Stage 2 from the initial pretrained model and *not* from the learned model from Stage 1, then the PGD steps in Stage 2 cannot converge. This means both the level of the noise injection and the initialization learned from Stage 1 are important for the success of Stage 2, showcasing the role of Stage 1 for the PAC-tuning approach.

### 5.2 The Necessity of Stage 2

To empirically verify the necessity of Stage 2 in PAC-tuning, we run PAC-tuning on the SST dataset

and validate how the training loss changes in the fine-tuning process. From Figure 3, it is clear that the training loss stagnates around 1.5 at the end of Stage 1 (200 epochs), which indicates that the model has not fit the data. This is because the existence of the $L_{PAC}$ term in the objective function prevents the optimizer from further decreasing $L_{train}$. As long as Stage 2 starts, the model quickly fits the data and the training loss is almost zero. More discussion on the two-stage training schema is available in Appendix B.

## 6 Advice for Applying PAC-tuning

In this section, we wish to share recommendations for using PAC-tuning, since the training process of PAC-tuning is different from conventional fine-tuning progress.

- In Stage 1, the target is to minimize $L_{train} + L_{PAC}$ which is larger than the training loss alone. Therefore, users may not observe a

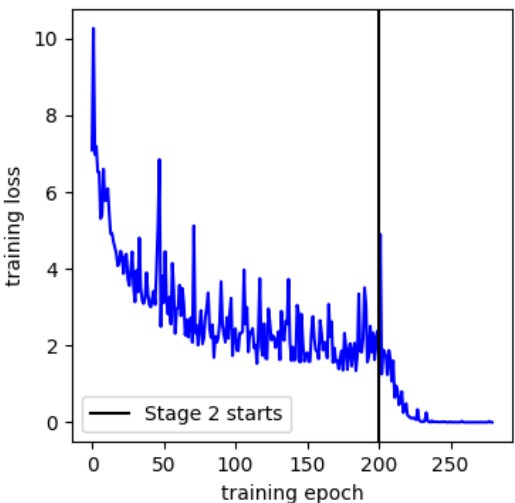

Figure 3: The training trajectory w.r.t. the training loss (cross-entropy loss) in the course of PAC-tuning with the SST dataset and BERT-base. We take 200 training epochs for Stage 1, and Stage 2 starts from the $200^{th}$ epoch as indicated by the black vertical line.

large decrease in the training loss. As long as the total loss is reducing, PAC-tuning is progressing correctly.

- Since the variances of the noises are non-negative, we used $\exp(2p)$ to model them, where $p$ is a trainable parameter. In addition, we initialize the standard deviation using the initial weights. More specifically, $p$ is initialized as the log magnitude of the initial weights.

- An effective way to check the status of Stage 1 is to check the mean value of the posterior variance. If the mean value does not change, users should increase their learning rate or increase the learning rate of the PLM.

- The prior variance parameters can easily approach convergence and are less sensitive to learning rate. Therefore, users can begin with a large learning rate and decrease it gradually. For the learning rate of the posterior variance, since the gradient is very small, we recommend readers to pick up a large learning rate.

## 7 Conclusions and Future Work

In this paper, we propose a PLM fine-tuning method, PAC-tuning, for few-shot text classification. PAC-tuning is based on PAC-Bayes training and perturbed gradient descent. We leverage PAC-tuning in the more challenging settings of larger models and smaller training data, which are

generally considered to be the two main obstacles for improving generalization through PAC-Bayes training. With extensive experiments on 5 GLUE benchmark tasks, we observed that the performance of PAC-tuning is competitive to other fine-tuning methods and more stable with respect to different model sizes and training dataset sizes. PAC-tuning, our proposed pretrained language model fine-tuning method, can be expanded in several ways for future work. More larger-sized models should be validated to fully explore the effectiveness of PAC-tuning. It would also be interesting to augment other fine-tuning techniques with PAC-tuning, especially data augmentation. Lastly, the performance of PAC-tuning is largely determined by the convergence of Stage 1, necessitating more studies to determine how to make Stage 1 converge quickly and robustly. Our experimental results demonstrate the usefulness of PAC-tuning and the potential to consider NLP problems from the point-of-view of generalization, a less explored PLM-optimization approach in the NLP community.

## 8 Limitations

Although we empirically validated the effectiveness of our proposed PAC-tuning method, there is still room for improvement. In particular, we cannot validate how PAC-tuning can be improved with the full-batch gradient update due to GPU hardware access limitations. Related to this, we did not perform an exhaustive best hyperparameter search, and instead defaulted to conventional learning rates and batch sizes to ensure fairness across all experiments.

It is also possible that the reported performances may not be the best performances. BERT and GPT-2 are not the newest language models and they are small compared to currently popular large language models. Therefore, more experiments for larger models are required, including experiments with close-sourced yet powerful models such as GPT-4. Furthermore, our experiments should be repeated in order to compare the performance of PAC-tuning and prompt-based techniques for validation against models such as ChatGPT and BARD.

## 9 Acknowledgement

This work is supported in part by the National Science Foundation (NSF) grant CCF-2212065.

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

## A   Hyperparameters

| Hyperparameters | Setting |
|:---:|:---:|
| **Optimizer** | AdamW |
| **Adam** $\beta_1$ | 0.9 |
| **Adam** $\beta_2$ | 0.98 |
| **Adam** $\epsilon$ | 1e-3 |
| **Learning rate for** $\theta$ | 5e-5 |
| **Learning rate for** $\omega$ | 1e-2 |
| **Maximum training epochs** | 35 |
| **Weight decay** | 0.01 |
| **Batch size** | 32 |

Table 5: Hyperparameter Settings for the AdamW Optimizer.

## B   Two-stage Approach

Most PAC-Bayes training methods typically rely on a single-stage approach. However, these methods are limited in their applicability, as they can only effectively handle bounded loss functions and shallow networks. They also struggle to optimize the noise prior, resulting in suboptimal final performance. We know of one other paper that used two stages but in a different way. More explicitly, Dziugaite and Roy (2017b) introduce a two-stage training process, where the first stage focuses on learning the model prior, followed by a second stage that learns the model posterior. Despite the use of two stages, as presented in Dziugaite and Roy (2017b), the method still faces challenges when dealing with unbounded loss functions, such as the commonly used cross-entropy loss in text classification tasks. Moreover, it demands a significant amount of training time.

To the best of our knowledge, prior to this work, there has been no PAC-Bayes training method that outperforms the baseline methods on any popular task, especially when targeting complex architectures like transformers.

The primary reason for the extended training epochs required in Stage 1 of PAC-tuning is the necessity to effectively learn both the model and the noise. It is worth noting that all PAC-Bayes training methods that optimize these aspects also tend to require more training epochs. While this does result in longer running times, the benefit of learned noise is significant, as it can be used to enhance model calibration and support pruning.