# OpenReview forum: "PAC-tuning: Fine-tuning Pre-trained Language Models with PAC-driven Perturbed Gradient Descent"
_EMNLP/2023/Conference — EMNLP 2023 Main_

### Official Review · Reviewer_uBJv · 2023-08-02

**Soundness:** 3

**Excitement:**

3: Ambivalent: It has merits (e.g., it reports state-of-the-art results, the idea is nice), but there are key weaknesses (e.g., it describes incremental work), and it can significantly benefit from another round of revision. However, I won't object to accepting it if my co-reviewers champion it.

**Missing References:**

Missing year in Haojie et al.

**Paper Topic And Main Contributions:**

This paper introduces PAC-tuning, a two-stage fine-tuning process for pre-trained language models. The authors focus on few-shot fine-tuning to showcase the applicability of PAC-based learning in challenging scenarios. The method shows competitive results on 5 tasks of the GLUE benchmark, performed using BERT and GPT-2.

**Questions For The Authors:**

- How do the parameter updates differ for each stage?
- How is each phase influencing the fine-tuning process? It would be nice to include the progress of the different losses during fine-tuning to help validate some claims.
- Did you consider evaluating PAC-tuning without the few-shot component? Would you still require more training than existing fine-tuning methods?

**Reasons To Accept:**

The paper focuses on exploring additional incorporations of PAC-based learning in deep neural networks, particularly large language models, which is a timely problem to tackle.

**Reasons To Reject:**

- The method requires a lengthy extra stage in the fine-tuning process instead of directly fitting the PLM classifier to the downstream task. Even though this might be okay in specific scenarios, it is still a major bottleneck in the method's practicality. I would like to see the authors attempt at fixing this problem. For example, an ablation study where Phase 2 uses random SGD perturbations is important to see the importance of  Phase 1 (or how it can be reduced).
- There are minimal to no discussions about the hyperparameter ranges tested and how the values for ξ and λ used in the experiments are obtained.
- Since the results on the 5 GLUE tasks are sometimes contradictory regarding the best method, extending the number of tasks should be considered to avoid noisy conclusions. The authors should also consider using SuperGLUE instead of GLUE.
- The GPT-2 GLUE scores are extremely low (less than 50%) which indicates that the problem setting was not well thought of. E.g. GPT-2 might not be a suitable model to perform fine-tuning in a few-shot learning setting.
- Finally, there are no results exploring the incorporation of PAC-tuning with other fine-tuning techniques such as pruning and data augmentation.

**Reproducibility:**

2: Would be hard pressed to reproduce the results. The contribution depends on data that are simply not available outside the author's institution or consortium; not enough details are provided.

**Reviewer Confidence:**

2: Willing to defend my evaluation, but it is fairly likely that I missed some details, didn't understand some central points, or can't be sure about the novelty of the work.

**Typos Grammar Style And Presentation Improvements:**

- Section 3.2,. -> line 315
- Several misuses of \citet vs \cite

---

> ### Author Rebuttal · Authors · 2023-08-28
>
> We thank the comments and feedback from the reviewer.
>
> **Q: The method requires a lengthy extra stage in the fine-tuning process instead of directly fitting the PLM classifier to the downstream task. Even though this might be okay in specific scenarios, it is still a major bottleneck in the method's practicality. I would like to see the authors attempt at fixing this problem. For example, an ablation study where Phase 2 uses random SGD perturbations is important to see the importance of Phase 1 (or how it can be reduced).**
>
> **A:** Our method does require extra training epochs as well as more running time, as shown in the table of our response to the reviewer **msWo**. Though PAC-tuning needs extra epochs, our method increases downstream performance with minimal overhead. And the test time of PAC-tuning is as less as the vallina-tuning. The extra training epochs is relevant to the learning rate of posterior noise variance, one of our future directions is to achieve adaptive learning rate by referring to [1]. On the other hand, the general PGD methods learn scaler noise but PAC-tuning generate parameter-wise noises that are informative to measure the generalization-wise importance.
>
> [1] Defazio, A., & Mishchenko, K. (2023). Learning-rate-free learning by D-adaptation. arXiv preprint arXiv:2301.07733.
>
> **Q: There are minimal to no discussions about the hyperparameter ranges tested and how the values for ξ and λ used in the experiments are obtained.**
>
> **A:** Regarding the initialization of noise variance parameters, the posterior noise variance is obtained with pre-trained weights’ mean values as described in Section 5. The prior noise variance is initialized as the mean of the posterior noise variance, and we will add this detail to the final version.
>
> **Q: Since the results on the 5 GLUE tasks are sometimes contradictory regarding the best method, extending the number of tasks should be considered to avoid noisy conclusions. The authors should also consider using SuperGLUE instead of GLUE.**
>
> **A:** Most baselines take the GLUE benchmark, allowing us to make comparisons. SuperGLUE is more suitable for more powerful language models, this was a better pairing considering we are using BERT.
>
> **Q: The GPT-2 GLUE scores are extremely low (less than 50%) which indicates that the problem setting was not well thought of. E.g. GPT-2 might not be a suitable model to perform fine-tuning in a few-shot learning setting.**
>
> **A:** GPT-family of models are all about few-shot learning, our results show their weaker performance in few-shot classification tasks. Given the weaker performance, PAC-tuning can still improve generalization, showing the effectiveness of our proposed method across modal architects.
>
> **Q: Finally, there are no results exploring the incorporation of PAC-tuning with other fine-tuning techniques such as pruning and data augmentation.**
>
> **A:** Our method can be easily extended to incorporate other off-the-shelf fine-tuning techniques. However, data augmentation would not be a good addition since it may introduce OoD samples contradicting the IID assumption in the PAC-bayes bound, since unseen words would be introduced with popular augmentation methods such as EDA and back-translation. We initially explored this addition for our paper and empirically validated this drawback. The only data augmentation method that meets the IID assumption is Mix-up, but augmentation with Mix-up has been shown to be problematic in determining correct labels. In the vision domain, PAC-tuning beyond Mix-up would be a good idea due to the continuous nature of imaging data. Lastly, the learned noise level in PAC-tuning is an indicator for parameter importance w.r.t generalization, and can be used for pruning. Our method can be seen as a counterpart to pruning.
>
> **Q: How do the parameter updates differ for each stage?**
>
> **A:** Figure 1 in our draft shows how the parameters in our method are updated. The parameters of the prior and posterior noise variances are learned in Stage 1, and the pre-trained and classification layers are updated in both Stage 1 and Stage 2 with the same learning rate.
>
> **Q: How is each phase influencing the fine-tuning process? It would be nice to include the progress of the different losses during fine-tuning to help validate some claims.**
>
> **A:** Section 3.2 describes this progress. We take the sentiment analysis task as an example and visualize the status of both cross-entropy loss (L_train term in Section 3.2) and the KL loss (L_pac term in Section 3.2). We apply 200 epochs for Stage 1. Please find the descriptions of Section 3.2 visualized in the [figure](https://www.dropbox.com/scl/fi/kar9kw8xmmg2lieqvwwel/ce_loss.png?rlkey=ogt4ght9mtimqjkyhxp027hsg&dl=0) for cross-entropy loss with and the [figure](https://www.dropbox.com/scl/fi/of6u9f5gk25hs7yk63vd5/kl_loss.png?rlkey=r9om4pxt8cy345d5g27ft6oik&dl=0) for KL loss
>
> **Q: Did you consider evaluating PAC-tuning without the few-shot component? Would you still require more training than existing fine-tuning methods?**
>
> **A:**  PAC-tuning works very well given a large number of training samples. Theoretically, a larger dataset size can make the L_pac loss term smaller, helping to minimize the L_train term and making the model fits training data better than that fine-tuned with only a few samples. We will add this in our final version as an analysis to the PAC-tuning.

---

### Official Review · Reviewer_6Piy · 2023-08-09

**Soundness:** 4

**Excitement:**

4: Strong: This paper deepens the understanding of some phenomenon or lowers the barriers to an existing research direction.

**Paper Topic And Main Contributions:**

In this study, the authors present a novel two-stage fine-tuning approach called PAC-tuning, aiming to enhance the generalization performance of downstream NLP tasks. In the first stage of PAC-tuning, the authors propose to learn the noise variance and update the model parameters by minimizing the PAC-Bayes bound. In the second stage, the noise variance learned in the first stage is utilized for noise injection to further fine-tune the model. The experimental results on the GLUE benchmark dataset demonstrate promising performance, particularly in small-sample text classification tasks. The paper is rich in content, well-supported by comprehensive experiments, and presents clear arguments. However, it lacks an in-depth analysis of the stability and convergence properties of PAC-tuning, as well as a more comprehensive comparison with other existing methods.

**Reasons To Accept:**

• Thorough and comprehensive content: The paper provides detailed explanations of the inspiration, background theory, method description, and experimental design of PAC-tuning. The overall structure of the paper exhibits clear and coherent reasoning.

• Strong practicality of the method: PAC-tuning can be easily applied to existing pre-trained language models. Experimental results demonstrate that PAC-tuning achieves superior performance compared to conventional fine-tuning and various other enhancement methods.

• Experimental results on the GLUE benchmark dataset show that PAC-tuning outperforms other methods, highlighting its effectiveness.

**Reasons To Reject:**

• In-depth analysis of the stability, convergence, and other performance aspects of PAC-tuning, along with a more comprehensive comparison with other fine-tuning methods, should be provided.

• It is necessary to investigate whether the two-stage approach of PAC-tuning introduces additional training time and computational overhead. Furthermore, a more extensive comparison with other methods should be included.

• More details regarding the implementation of PAC-tuning and the exploration of key hyperparameter settings should be provided.

• The computational complexity and the curse of dimensionality associated with PAC-Bayes bounds should be discussed. PAC-tuning currently considers only the parameter confidence differences between the pre-trained language model (PLM) and the classification layer, neglecting differences among different layers. The discussion could be expanded to include: 1) Confidence differences among different layers of the PLM, and 2) The computation complexity of PAC-Bayes bounds and the impact of the curse of dimensionality, along with potential mitigation strategies.

• A comparison with current state-of-the-art prompt-tuning methods is lacking. If GPT-3 and GPT-4 are considered too powerful as baselines, alternative language models of different scales, such as BERT-large or DistilBERT, should be used to validate the scalability of the proposed method. The effectiveness of PAC-tuning needs further verification beyond experiments conducted solely on BERT-base.

**Reproducibility:**

4: Could mostly reproduce the results, but there may be some variation because of sample variance or minor variations in their interpretation of the protocol or method.

**Reviewer Confidence:**

4: Quite sure. I tried to check the important points carefully. It's unlikely, though conceivable, that I missed something that should affect my ratings.

---

> ### Author Rebuttal · Authors · 2023-08-28
>
> Thanks for the helpful comments. Below our responses to the concerns above.
>
> **Q: In-depth analysis of the stability, convergence, and other performance aspects of PAC-tuning, along with a more comprehensive comparison with other fine-tuning methods, should be provided.**
>
> **A:** Compared to traditional training, our proposed PAC-tuning approach utilizes a different objective function and an extra parameter to model the noise level. But the optimization algorithm we used to solve this new optimization problem is still Adam, therefore, the convergence property and stability follow those of Adam’s directly.  We will add the training trajectory of PAC-tuning and other methods for more comprehensive comparison.
>
>
> **Q: It is necessary to investigate whether the two-stage approach of PAC-tuning introduces additional training time and computational overhead. Furthermore, a more extensive comparison with other methods should be included.**
>
> **A:** The first stage in the two-stage approach indeed increases the running time for a single run. But since it avoids hyper-parameter tuning, the overall running time of the PAC-tuning is similar to other methods as shown in the table in our response to the reviewer **msWo**. Even though the PAC-tuning introduces the noise level as extra parameters, we still only need one back-propagation in each iteration, as the result of this one back-propagation can be used to compute the gradients of both the model and the noise. Although we implement stage 1 with 250 epoch for all tasks in our reported experiments, for some tasks the training epochs for stage 1 can be very small such as sentiment analysis, due to the less difficulty of this task.if one hopes to shorten the computational time, then some tuning may be needed on the learning rate, as a large learning rate grants a faster training convergence.  But in the future we plan to use the method  [1] to automatically tune the learning rate. We would  add a short  discussion on this in the final version.
>
>
> [1] Defazio, A., & Mishchenko, K. (2023). Learning-rate-free learning by D-adaptation. arXiv preprint arXiv:2301.07733.
>
> **Q: More details regarding the implementation of PAC-tuning and the exploration of key hyperparameter settings should be provided.**
>
> **A:** We will open-source our code and add more details about how to compute hyperparameters, e.g. K,  in section 3.5, upon acceptance.
>
> **Q:  The computational complexity and the curse of dimensionality associated with PAC-Bayes bounds should be discussed. PAC-tuning currently considers only the parameter confidence differences between the pre-trained language model (PLM) and the classification layer, neglecting differences among different layers. The discussion could be expanded to include: 1) Confidence differences among different layers of the PLM, and 2) The computation complexity of PAC-Bayes bounds and the impact of the curse of dimensionality, along with potential mitigation strategies.**
>
> **A:** (1) Regarding the confidence with respect to different layers among the pre-trained part or the classification part: we did layer-wise prior parametrization and elementwise posteriror parameterization，which means, difference between layers are taken into consideration. We will add these details in the final version.  (2) The PAC-turning algorithm is almost as expensive as the traditional training. Even though the PAC-tuning introduces the noise level as extra parameters, we still only need one back-propagation in each iteration, as the result of this one back-propagation can be used to compute the gradients of both the model and the noise levels.  Therefore, the complexity of the optimization algorithm does not suffer from the curse of dimensionality. From our experiment, the PAC-Bayes bound also does not seem to suffer from the curse of dimensionality in the sense that the numerical value of the bound does not seem to grow with the number of model parameters.  This would be an interesting question to study.
>
>
> **Q: A comparison with current state-of-the-art prompt-tuning methods is lacking. If GPT-3 and GPT-4 are considered too powerful as baselines, alternative language models of different scales, such as BERT-large or DistilBERT, should be used to validate the scalability of the proposed method. The effectiveness of PAC-tuning needs further verification beyond experiments conducted solely on BERT-base.**
>
> **A:** The Prefix-tuning baseline in our paper is the referenced baseline methods of prompt-tuning with Bert-base/large and GPT-2.  We reported the results of PAC-tuning on BERT-large in Table 4.

---

### Official Review · Reviewer_msWo · 2023-08-11

**Soundness:** 4

**Excitement:**

4: Strong: This paper deepens the understanding of some phenomenon or lowers the barriers to an existing research direction.

**Paper Topic And Main Contributions:**

This paper presents a novel few-shot finetuning technique for pre-trained LMs. They introduce the PAC training strategy to model fine-tuning and obtain a remarkable improvement compared to previous fine-tuning approaches.

**Questions For The Authors:**

Please respond to the concerns in weaknesses.

**Reasons To Accept:**

PAC-tuning presented in this paper takes the first step to applying the PAC training to LM fine-tuning. It reveals an avenue of controllably tuning the large pre-trained LMs with few-shot samples. And the reported improvements on standard benchmarks compared to existing model tuning approaches are remarkable.

**Reasons To Reject:**

Weaknesses:

A. Although the proposed two-stage finetuning method brings clear improvements compared to previous tuning methods, I'm curious about the performance of the tuned model after the first stage. And it's better to explain further why the two-stage design is indispensable instead of a collaborative optimization.

B. It is advisable to provide a comparison of the number of tunable parameters and the amount of tuning time budgets between the listed tuning methods in the experiment.

C. The foreshadowing of introducing PAC training is poor. It's better to explain the minimization of the PAC-Bayes bound solves what problem in the current fine-tuning landscape.

D. The current experiment only involves the NLU task, and testing on the NLG task would be useful.

**Reproducibility:**

3: Could reproduce the results with some difficulty. The settings of parameters are underspecified or subjectively determined; the training/evaluation data are not widely available.

**Reviewer Confidence:**

3: Pretty sure, but there's a chance I missed something. Although I have a good feel for this area in general, I did not carefully check the paper's details, e.g., the math, experimental design, or novelty.

---

> ### Author Rebuttal · Authors · 2023-08-28
>
> We thanks for the reviewer's comments.
>
> **Q: Although the proposed two-stage finetuning method brings clear improvements compared to previous tuning methods, I'm curious about the performance of the tuned model after the first stage. And it's better to explain further why the two-stage design is indispensable instead of a collaborative optimization.**
>
> **A:** In conclusion stage 1 is necessary to determine the learned noise and help the model to be in a good initialization point for continual fine-tuning. Stage 2 is necessary to minimize training loss as much as possible while avoiding under- and overfitting. This [figure](https://www.dropbox.com/scl/fi/kar9kw8xmmg2lieqvwwel/ce_loss.png?rlkey=ogt4ght9mtimqjkyhxp027hsg&dl=0) shows that the model underfits training data before stage 1, and well fit the training data during stage 2. To understand the effect on the model initialization point, this [figure](https://www.dropbox.com/scl/fi/dowt6xx96ck03zjirqzlw/trainimprovestage1.png?rlkey=l00o4qqmve1mye6pwoufrvwsw&dl=0) shows that if we directly inject the learned noise into the pretrained model without stage 1, the training loss diverges. It would be very great if we can do collaborative optimization over the model and noise, the underfitting in stage 1 is inevitable because of the existence of the PAC loss.
>
> **Q: It is advisable to provide a comparison of the number of tunable parameters and the amount of tuning time budgets between the listed tuning methods in the experiment.**
>
> **A:** One of the benefits of the proposed algorithm is easy tuning.  In contrast to the Adam baseline, the final performance of the proposed method depends very little on the hyper-parameters.  In particular, the proposed method does not need dropout, and the generalization is not affected much by varying the learning rates, as long as the learning rate is not too large to cause it to diverge. Likewise,  for the proposed method, using large batch sizes or small batch sizes  yield similar generalization result.  Having said that, if one hopes to shorten the computational time, then some tuning may be needed on the learning rate, as a large learning rate grants a faster training convergence.  But in the future we plan to use the method  [1] to automatically tune the learning rate.We would  add a short  discussion on this in the final version.
> We have provided the table below to answer your question, and will include this in the final version.
> In  the table below, we  compare the running time with  those of vanilla-tuning, data augmentation, and noise injection on the SST dataset. Since  parameter-efficient approaches are not competitive to those methods, we did not report the running time for parameter-efficient approaches.  All experiments (except PCA) take 15 training epochs over 5 seeds, and we run PAC-tuning for  280 epochs (we can further reduce the epochs for Stage 1 for most of the tasks). Though PAC-tuning uses more epochs, we just need to run it once, as opposed to having to run multiple times of other methods for hyper-parameter tuning. Therefore, the running times of all the methods are comparable.
>
>
>
>
> | Method            | Hyper-parameter setting                                                                                          | Running time  |
> |-------------------|------------------------------------------------------------------------------------------------------------------|---------------|
> | Vanilla-tuning    | batch_sizes: [8, 16, 32, 64] Learning rate:[1e-5, 2e-5, 3e-5, 5e-5]                                              | 19.8 minutes  |
> | Data augmentation | batch_sizes: [8, 16, 32, 64] Learning rate:[1e-5, 2e-5, 3e-5, 5e-5]                                              | 38.8 minutes  |
> | Noise Injection   | batch_sizes: [8, 16, 32, 64] Learning rate:[1e-5, 2e-5, 3e-5, 5e-5] noise_level = [1e-8, 1e-7, 1e-6, 1e-5, 1e-3] | 26.8minutes   |
> | PAC-tuning        | Batch size: [32] Learning rate: [5e-5] Learning rate for posterior noise variance:[0.01]                         | 35.25 minutes |
>
> **Q: The foreshadowing of introducing PAC training is poor. It's better to explain the minimization of the PAC-Bayes bound solves what problem in the current fine-tuning landscape.**
>
> **A:** Upon acceptance, we will add more details about how minimizing pac-bayes bound helps to minimize the generalization/test error   on the fine-tuning task.
>
> **Q: The current experiment only involves the NLU task, and testing on the NLG task would be useful.**
>
> **A:** We feel there is no technical obstacle in applying the proposed method on NLG.  However, good generalization is only one of the desired properties for NLG. High-level evaluation metrics like coherence, harmless or model-based evaluation metrics such as Bert-score are also very important. How to incorporate non-generalization evaluation into the PAC-Bayes training is one of our feature directions.

---

### Official Review · Reviewer_NWbh · 2023-08-14

**Soundness:** 3

**Excitement:**

4: Strong: This paper deepens the understanding of some phenomenon or lowers the barriers to an existing research direction.

**Paper Topic And Main Contributions:**

This paper introduces a novel fine-tuning technique for pre-trained language models (PLM) aimed at enhancing generalization performance. Termed "PAC training," the method seeks to directly minimize the PAC-Bayes generalization bound and is executed in two distinct stages. Initially, the method determines the optimal parameter distribution variances. Subsequently, the gradient is adjusted by injecting noise, leveraging the variance ascertained in the first phase. Through rigorous experimentation, the paper demonstrates that PAC tuning surpasses existing fine-tuning methods in terms of generalization performance.

**Questions For The Authors:**

Is the proposed method related to Bayesian neural networks (BNNs)? In BNNs, we also model the parameter as a distribution. The PGD part  is very similar to BNNs which is inferred using stochastic gradient langevin dynamics. Given this, I am very curious about whether the proposed method can improve uncertainty estimation and model calibration?

**Reasons To Accept:**

1. The paper is cogently composed with a lucidly articulated motivation.
2. The proposed methodology is technically robust. Employing PGD to minimize the PAC-Bayes bound emerges as an astute and efficacious approach.

**Reasons To Reject:**

The chosen baselines in the experimental section appear somewhat unconvincing. Many are geared towards efficiency rather than improved generalization performance. Numerous fine-tuning techniques have been developed with an emphasis on bolstering generalization performance. How does the PAC training method stack up against the likes of SMART?

SMART: Robust and Efficient Fine-Tuning for Pre-trained Natural Language Models through Principled Regularized Optimization.

**Reproducibility:**

4: Could mostly reproduce the results, but there may be some variation because of sample variance or minor variations in their interpretation of the protocol or method.

**Reviewer Confidence:**

4: Quite sure. I tried to check the important points carefully. It's unlikely, though conceivable, that I missed something that should affect my ratings.

---

> ### Author Rebuttal · Authors · 2023-08-28
>
> We appreciate the great comments from the reviewer.  Here are our responses to the proposed questions.
>
> **Q: The chosen baselines in the experimental section appear somewhat unconvincing. Many are geared towards efficiency rather than improved generalization performance. Numerous fine-tuning techniques have been developed with an emphasis on bolstering generalization performance. How does the PAC training method stack up against the likes of SMART?**
>
> **A:** We did implement tests using SMART. Following the suggested hyperparameter settings, SMART performed lower than our vanilla-tuning baseline for most tasks. One possible cause could be the limited training dataset size hindering the search for adversarial samples, resulting in a less effective decision boundary of smoothness. Because of this, we did not report our SMART results, however we can add them into the final version with this explanation. Data augmentation is a widely used method for improving generalization and robustness by introducing more samples with off-the-shelf tools. Noise injection is considered a regularization over the trace of hessian matrix, improving generalization.
>
> **Q: Is the proposed method related to Bayesian neural networks (BNNs)? In BNNs, we also model the parameter as a distribution. The PGD part is very similar to BNNs which is inferred using stochastic gradient langevin dynamics. Given this, I am very curious about whether the proposed method can improve uncertainty estimation and model calibration?**
>
> **A:** As you correctly mentioned, BNN and PAC-bayes training both model the parameter as a distribution and they both target on mitigating overfitting and improving generalization. The differences are 1) PAC-Bayes training directly minimizes the generalization error while the relation between generalization the BNN minimizer is a bit vague 2) During training, BNN uses a fixed prior, while our PAC-bayes training algorithm optimizes over a set of priors for the best one, therefore is tighter 3) During the test stage, our PAC-bayes training algorithm uses a fast deterministic predictor while BNN algorithms use probabilistic predictors.  These three differences all contribute to the reported improvement in the numerical results. For your  second question, yes, since PAC-Bayes training also learns the distribution, we do expect to apply it to uncertainty estimation and model calibration in  our future work.

---

### Meta-Review · Area_Chair_bNMm · 2023-09-18

**Recommendation:** 4

**Metareview:**

The author propose a hypeparameter optimization + gradient descent modification method which is inspired by minimizing a generalization bound. All reviewers agree that the synthesis of PAC-bayes generalization bounds with PLMs is interesting, and while there are some shared concerns about baselines and evaluations (to other regularized fine tuning methods, prompting etc), this is not a major issue.

---

### Decision · Program_Chairs · 2023-10-07

**Decision:**

Accept-Main

**Comment:**

The author propose a hypeparameter optimization + gradient descent modification method which is inspired by minimizing a generalization bound. All reviewers agree that the synthesis of PAC-bayes generalization bounds with PLMs is interesting, and while there are some shared concerns about baselines and evaluations (to other regularized fine tuning methods, prompting etc), this is not a major issue.